# Interactive biocatalysis achieved by driving enzyme cascades inside a porous conducting material
Bhavin Siritanaratkul [1], Clare F. Megarity [2], Ryan A. Herold[3,4] & Fraser A. Armstrong [3] ✉

An emerging concept and platform, the electrochemical Leaf (e-Leaf), offers a radical change in the way tandem (multi-step) catalysis by enzyme cascades is studied and exploited. The various enzymes are loaded into an electronically conducting porous material composed of metallic oxide nanoparticles, where they achieve high concentration and crowding – in the latter respect the environment resembles that found in living cells. By exploiting efficient electron tunneling between the nanoparticles and one of the enzymes, the e-Leaf enables the user to interact directly with complex networks, rendering simultaneous the abilities to energise, control and observe catalysis. Because dispersion of intermediates is physically suppressed, the output of the cascade – the rate of flow of chemical steps and information – is delivered in real time as electrical current. Myriad enzymes of all major classes now become effectively electroactive in a technology that offers scalability between micro-(analytical, multiplex) and macro-(synthesis) levels. This Perspective describes how the e-Leaf was discovered, the steps in its development so far, and the outlook for future research and applications.

Biocatalysis—a sector of great importance for synthesis, and for enzyme or drug development by university laboratories and chemical/pharmaceutical industries—is increasingly making use of enzyme cascades, in which several enzymes are combined to catalyse a multi-step process[1–6]. Instead of being dispersed in solution, enzyme cascades can be immobilised on materials or entrapped within scaffolds or nanocontainers[7–18], thereby fixing the catalysts in a practically advantageous configuration and (often) a well-defined environment. However, in general, and whether immobilised or otherwise, the operation of enzyme cascades is noninteractive and passive, as processes are normally initiated by adding reactants (or irradiating) and monitored by analysing product release. A new and fecund direction, the Electrochemical Leaf (e-Leaf) allows the user to manipulate and exploit enzyme cascades in an interactive manner with ease and precision[19], enabling *dialogue* in terms of simultaneous energization (*input*), control (*moderation of input*) and real-time observation (*output*). The concept and practice draw on a wide breadth of disciplines, embracing both physical and biological sciences. Representing a significant advance in protein film electrochemistry (PFE)[20,21], the e-Leaf induces and records the electrical current arising from sequential chemical reactions catalysed and tightly channelled by nanoconfined enzyme cascades. Enzymes of all classes can be integrated and, as explained in this

Perspective, rendered effectively 'electroactive': they do not need to possess a redox-active centre.

## Achieving enzyme nanoconfinement with a simple material

An underlying theme of the e-Leaf is the nanoconfined environment afforded by a simple procedure in which hydrophilic, electronically-conducting nanoparticles of a metallic oxide (MO) are coated on a conducting support by electrophoretic deposition to give a robust mesoporous layer (Fig. 1a)[22–24].

The electrophoretic deposition method is rapid and simple to use. A useful analogy is to consider applying a little mortar (representing the attractive intercolloidal interactions[24]) to individual house bricks and throwing them onto a road: rather than a neat wall, the bricks form a pile with gaps and tunnels similar in dimensions to the bricks themselves, but with random distribution (Fig. 1b). Instead of the sizes of the nanoparticles themselves being of primary importance, it is the spaces created between them that are directly relevant, as these spaces will take up and entrap enzyme molecules after they are deposited on the surface of the coating (Fig. 1c). The term 'nano-' is used to describe the enzyme's environment (rather than 'meso-' which is usually taken to describe the overall porosity

[1]Department of Chemistry, University of Liverpool, Liverpool L69 7ZF, UK. [2]Department of Chemistry, Manchester Institute of Biotechnology, University of Manchester, Manchester M1 7DN, UK. [3]Department of Chemistry, University of Oxford, Oxford OX1 3QR, UK. [4]Department of Chemistry and Biochemistry, University of California, San Diego, La Jolla, CA 92093, USA. ✉e-mail: fraser.armstrong@chem.ox.ac.uk

**Fig. 1 | The cascade enzymes are loaded into a porous electrode material. a** SEM image of a layer of indium tin oxide nanoparticles (size distribution 10–50 nm) deposited on indium tin oxide glass. **b** A helpful metaphor: bricks thrown into a pile create random spaces having dimensions relating in size to the bricks. **c** Enzymes deposited onto the surface enter the pores and permeate throughout the porous layer. The image at the right shows an impression of magnified pores crowded with enzymes (ferredoxin-NADP$^+$ reductase, mean diameter ~55 Å, and two other enzymes, mean diameters 108 Å and 160 Å); enzymes and pores are to scale.

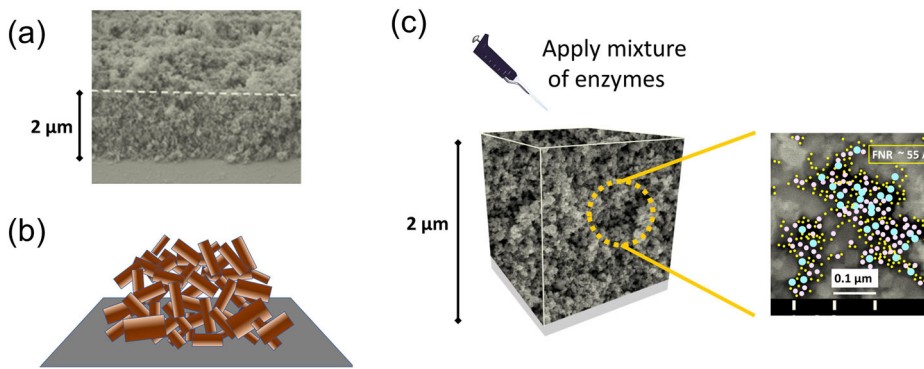

**Fig. 2 | Stages in the discovery of the e-Leaf.**
**a** Cyclic voltammogram showing the reduction and reoxidation of the FAD cofactor in FNR when it is embedded in a porous ITO electrode. The result shown refers to pH 8.0. The narrow widths of oxidation and reduction peaks signified a two-electron transfer process having significant cooperativity and very little dispersion[21,30] despite the unusual environment. **b** Achieving the reversible catalytic electrochemistry of NADP$^+$/NADPH (red cyclic voltammogram, overlaid with the 'non-turnover' FNR signal shown in black). **c** Activation of cascade electrocatalysis following pre-loading of FNR and then injecting different dehydrogenases (to low concentration) into the cell solution, which also contained the substrate for the enzyme being studied. ADH alcohol dehydrogenase, RedAm reductive aminase, ME malic enzyme (malate dehydrogenase), (S)-IRED (S)-imine reductase.
**d** Cyclic voltammograms for the electrocatalysis of ketone/alcohol interconversion: at pH 9.0 the rate is almost the same in each direction. Figures were adapted with permission: (a and b)[19], (c)[38] and (d)[39].

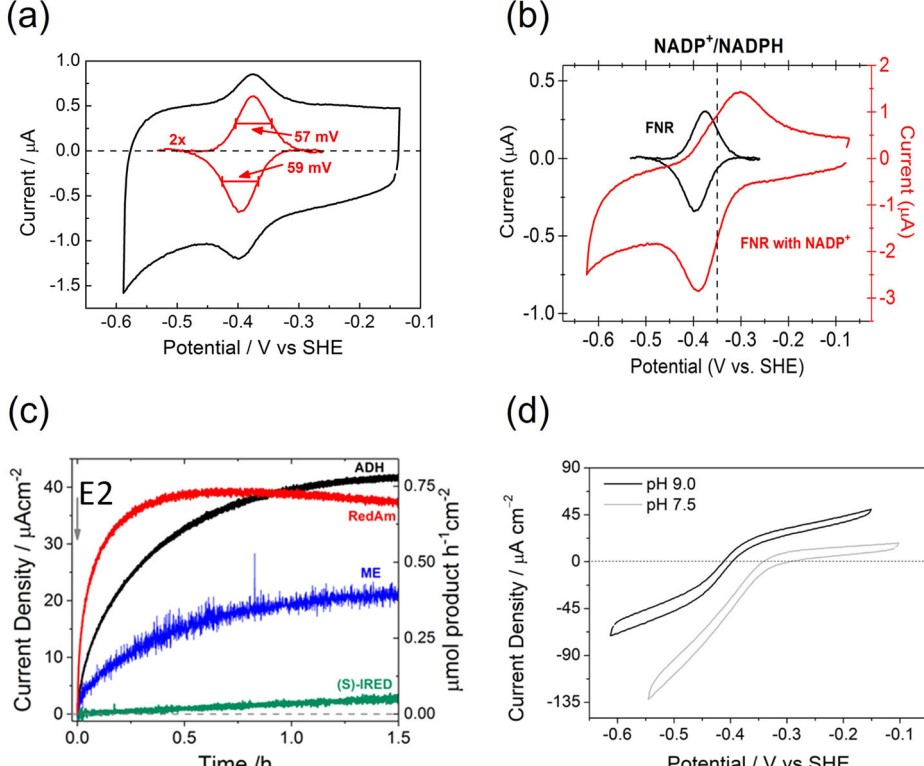

scale) as it is self-explanatory and widely recognised across different scientific disciplines: it also emphasises the restricted space surrounding enzyme molecules. The random arrangement of pore dimensions is probably advantageous as it allows enzymes having a wide range of sizes to be trapped[19]. Despite the obvious fact that the inorganic material appears far removed from any recognisable biological counterpart, its ability to accommodate the many different enzymes required for a specific catalytic process in a very crowded fashion mimics properties of biological compartments in cells[25]. Indeed, with the additional ability to supply electrical energy, the trapped enzymes do effectively 'come to life'.

## Development of the e-Leaf
Electrodes comprised of conducting mesoporous MO layers have been used for many years in electrochemical investigations, including studies of single redox-active proteins—cytochrome c being a notable example[26]. The conventional focus of interest has been the ability to achieve much greater coverage of redox-active material compared to that possible with a planar electrode. The e-Leaf began[27] with the discovery of the reversible electrochemistry of ferredoxin-NADP$^+$ reductase (FNR), the small flavoenzyme

(39 kDa) responsible for channelling electrons energised by photosynthesis into the Calvin–Benson–Bassham cycle through which atmospheric CO$_2$ is incorporated into organic molecules[28,29]. Several important observations were made upon introducing FNR to a solution in which a mesoporous indium tin oxide (ITO) electrode (formed by electrophoretic deposition of a layer of nanoparticles on a pyrolytic graphite edge electrode) was immersed. A pair of oxidation and reduction peaks (a voltammetric signal) characteristic of an immobilised redox couple appeared at the electrode potential expected for the tightly-bound FAD (flavin adenine dinucleotide) cofactor (Fig. 2a). The narrow line shape was as expected[30] for a two-electron transfer having a high (but not total) degree of cooperativity, and the small separation between oxidation and reduction peaks implied fast electron exchange. In the ideal reversible case, the cyclic voltammogram for a surface-immobilised redox couple consists of a pair of Gaussian-like peaks corresponding to reduction and oxidation—each maximising at the same electrode potential and (provided all the molecules experience the same local environment) displaying a half-height width of approximately $90/n_{eff}$ mV at 25 °C. The term $n_{eff}$ is the 'effective electron number'—the number of electrons transferred in an apparently simultaneous (cooperative) process[31].

For a single electron transfer, $n_{eff}$ is automatically equal to 1.0, but for a two-electron reaction $n_{eff}$ depends on the separation between the component one-electron potentials. If the second electron transfers at a much more favourable potential than the first ($E_2 \gg E_1$), the one-electron intermediate is very unstable and $n_{eff} = 2$ (the limiting case for a fully cooperative two-electron transfer). The peak height varies as $n_{eff}^2$, further making such a cooperative transfer much easier to observe. The partially cooperative nature of the FNR signal ($1 < n_{eff} < 2$) was significant because, in the chloroplast, FNR receives two sequential one-electron transfers from separate ferredoxin molecules—requiring the one-electron radical to have sufficient inherent stability to exist as an intermediate[28,29,32,33]. Very significantly, the signal intensity was far higher than could be expected from a monolayer, meaning that the FNR must be trapped deeply in the pores, yet remain monodisperse (each enzyme molecule thus experiencing a similar local environment)[30]. The quantity of FNR detectable by its electroactivity varied with pH, increasing to 500 picomoles cm$^{-2}$ (equivalent to 100 monolayers) at pH 9[27].

The term 'electrochemical leaf' (e-Leaf) was coined once it was realised that the FNR (observed initially as a 'non-turnover' signal) catalyses the rapid interconversion between NADP$^+$ and NADPH when either of these is added to the solution, thus mimicking its role in photosynthesis (Fig. 2b)[28,29]. But unlike a real leaf, a human operator could now control the reactions in both directions, literally at the touch of a laptop key. That the electrocatalysis of NADP$^+$/NADPH interconversion by FNR was not only bidirectional but *reversible* would prove to be significant for the technology. Many enzymes are now known to be reversible electrocatalysts when attached to an electrode[34,35]. The term 'reversible' is often used casually to describe a reaction that is simply bidirectional, i.e. can be driven in either direction: the electrochemical (thermodynamic) definition of reversibility is far stricter—it refers to a bidirectional reaction for which the direction and rate respond fluently to a miniscule change in potential across the formal potential value, i.e. only a very small overpotential (driving force) is needed[34,36]. Electrochemical reversibility is a marker for efficiency—it signifies that the energy input is used to conserve thermodynamic requirements rather than overcome kinetic barriers. The reversibility extended to the NADP$^+$/NADPH catalytic recycling process—notably, the fine-tuning of rate and direction within a modest potential range would make it possible to control processes in the most delicate way.

Viewed in more fundamental terms, the discovery that FNR behaves as a reversible electrocatalyst for NADP$^+$/NADPH interconversion came as no surprise, given the enzyme's role of transducing solar-driven electrical and chemical energy in photosynthesis[28,29]. Indeed, the need to minimise overpotential among enzymes of a redox chain must have been an important (but unusual) evolutionary driver[35]. It was also significant that the electrochemistry of FNR (Fig. 2a) was stable and almost textbook in its appearance[30], despite the FAD cofactor being exposed and non-covalently bound in a small enzyme (<40 kDa) along with the requirement, for catalytic function, that it bind and release a NADP(H) molecule once for every two electron transfers[28,33,37]. As explained later, the slightly more negative reduction potential of the FAD cofactor relative to NADP$^+$ gave a simple (albeit superficial) explanation for why the peak due to catalytic reduction of NADP$^+$ (Fig. 2b) is sharper than the corresponding oxidation peak (reduced FAD is a better reductant whereas oxidised FAD is a weaker oxidant). As would eventually become clear, NADP$^+$ and NADPH cofactors are largely contained within the electrode pores, and the voltammetry shown in Fig. 2b resulted from pore-restricted rather than semi-infinite diffusion of the mobile cofactor.

The second stage of the discovery took place once it was proved that the coupling of FNR-catalysed NADP(H) recycling to a dehydrogenase—one catalysing hydride transfer to other molecules—required both enzymes to be incorporated into the electrode pores[38]. This requirement was established by introducing a dehydrogenase at low concentration into the solution surrounding an electrode that had already been loaded with FNR in the ITO pores. In electrocatalysis, the current is directly proportional to the rate that the reactant is processed: in this case, the catalytic current due to the reaction

characteristic for each particular dehydrogenase increased with time, i.e. coupled catalytic activity between FNR and the dehydrogenase started and rose as the dehydrogenase entered the pores (Fig. 2c). Proof of the redundancy of dehydrogenase molecules remaining in solution was obtained by exhaustive solution exchange which did not decrease the current. Anticipating further progress, a generic shorthand description was introduced—(E1 + E2 + …)@MO/support; where E1 represents FNR (the transducing enzyme), E2 is a dehydrogenase, and MO/support is a porous layer of conducting metal oxide deposited on a support that could range from micro- to macro-scale in dimensions. The inclusion in the title 'Electrocatalytic Volleyball…' of the first paper in *Angewandte Chemie* alluded to the likelihood that NADP(H) molecules in the pores must be rapidly passed back and forth between all enzymes (E1 ↔ E2, E1 ↔ E1, E2 ↔ E2), with confinement increasing the encounter frequency and the probability that productive hydride transfer ultimately takes place. The resulting electrochemistry of an alcohol dehydrogenase catalysing alcohol/ketone interconversion is shown in Fig. 2d, where it is seen that the current represents the direction and rate at each electrode potential[39]. The reaction is bidirectional at pH 9.0, but not at pH 7.5.

It would soon become clear that further enzymes beyond the dehydrogenase could be incorporated, extending the chain and allowing the investigation of complex cascades in an interactive and dynamic fashion. The entry of enzyme molecules into the pores appears to be spontaneous, likely factors being that each macromolecule displaces a substantial quantity of ordered water molecules (entropic), along with interactions between the irregular enzyme surface and the internal nanoparticle surfaces. It is known that enzyme molecules are stabilised when entrapped in porous materials[40], and in this case, the polar, hydrophilic nature of ITO and other conducting metal oxides must be very relevant. The enzymes become highly concentrated (often millimolar) implicating crowding on a scale resembling that in living cells[25]. Whereas the quantity of FNR could be determined directly from the integrated area of the non-turnover FAD peaks; as mentioned later, the use of a dehydrogenase carrying its own tightly bound cargo of NADP(H) into the pores would allow the quantity of E2 to be measured also[41].

The fact that coupled FNR-E2 electrocatalysis becomes bidirectional under certain conditions emphasises two points. First, if the ketone and alcohol (or the components of any other redox couple) are present in the solution at equal concentrations, the potential at which the voltammetric trace crosses the zero-current axis is the formal reduction potential for the ketone/alcohol half-cell reaction (see trace at pH 9.0 in Fig. 2d). Although the electrons are transferred via NADP(H) cycling, the overall reaction being observed is the reduction or oxidation of the other substrates for E2, provided the concentration of NADP(H) is much lower than the reactant (i.e. it is present only in catalytic amounts). Second, in order for the coupled FNR-E2 to appear bidirectional, the formal potential of the reaction catalysed by E2 must lie quite close to that of the NADP$^+$/NADPH couple (−0.32 V at pH 7.0, −0.35 V at pH 8.0, etc). In this situation, the NADP(H) is analogous to the 'electrochemical control centre' referred to in PFE and which is normally an internal electron-relay centre or active site in the case of a single redox enzyme[42–44]. As explained later, an important factor determining the preferred direction (the catalytic bias) of the overall electrocatalytic reaction is the difference between the formal reduction potential of the NADP$^+$/NADPH couple and the E2 substrate/product couple—a principle that also underpins the selection of suitable electron mediators for potentiometric titrations[43,45]. In Fig. 2d, the formal potential of the NADP$^+$/NADPH couple (pH dependence −30 mV/pH unit) is closely matched with the ketone/alcohol couple (−60 mV/pH unit) at pH 9.0, but at pH 7.5 it is relatively too negative to mediate alcohol oxidation.

From that point on (2019), there was confidence that it would be feasible to design devices able to engage a host of enzyme cascades interactively, extending the scope to systems comprised of several different enzymes catalysing linear or branched sequences of reactions. The emphasis would soon go well beyond cofactor recycling, the latter having been a topic of considerable interest and importance for many years[46,47]. As discussed

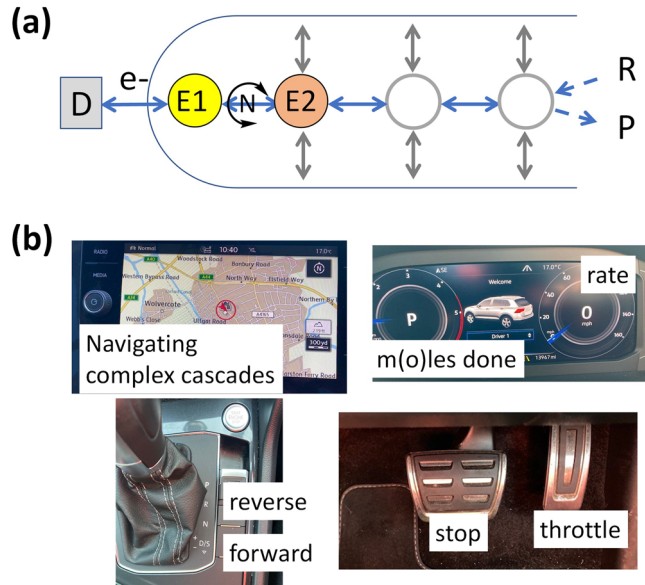

**Fig. 3 | Useful metaphoric representations of the e-Leaf. a** As an urban transport map in which the enzymes (stations) are arranged in terms of next-neighbour in the catalytic sequence noting (as is obvious for transport) there is a minimum requirement for two 'stations', in this case E1 (FNR) and E2 (an NAD(P)(H)-dependent dehydrogenase); N, R and P represent NAD(P)(H), reactant and product, respectively; D (the dashboard) comprises the electrochemical workstation and other hardware. **b** The dashboard of a car, emphasising the various controls and displays that have equivalents in the e-Leaf.

later, the E1–E2 pair is the essential minimal component, E1 for transduction and E2 for coupling cofactor recycling to the reaction of interest; loading E1 alone would simply use more enzyme with no advantage gained[48]. Thus far, E1 has been almost exclusively an N-terminal His-tagged recombinant version of FNR from *Chlamydomonas reinhardtii*, which favours NADP(H), but in some experiments, the FNR Y354S variant has been used, for which the catalytic preference is switched to NAD(H)[49,50].

It is useful to metaphorize the e-Leaf by reference to two familiar themes. The first of these themes (Fig. 3a) represents a confined cascade system in terms of an urban transport map, in which the enzymes are organised along a line spanning two boundaries—the electron-tunnelling process controlled by the *Dashboard* D (left) and the entry and escape of reactants (R) and products (P) at the pore entrance (right).

The map depicts the minimal functional pair E1 + E2 and options for extending the cascade to include further enzymes; E3, E4 and so on. It is expected, since enzymes are present at high concentrations and the total depth of a typical MO layer is <10 microns, that the distances between enzymes will always be short enough that diffusion of small reactants is not the rate-determining factor, i.e. it does not limit the catalytic current[51]. Thus, two enzymes located adjacently on the map do not need to be in very close physical proximity[52]; rather, it is more important that a molecule transferring between them does not escape. For instance, under nanoconfinement in the finite depth of the MO layer, it is only important that enzyme E3 is the 'neighbouring station' of enzyme E2 in terms of its position in the reaction sequence—physical separation over such a short range being less important. Enzyme-catalysed reactions are highly specific and usually so efficient that the product of one enzyme will be processed by the next enzyme before it escapes. In the packed pore environment, the term 'cluster channelling' is highly relevant[53]. The cascade map stresses the *information flow* transduced by E1 into current, and it standardises e-Leaf representations; connections are easily introduced, showing where branches or competing routes join.

The second theme (Fig. 3b) concerns the Dashboard and the way that it now becomes easy to control reaction direction and rate, as well as navigate across complex cascades, consolidating *input* and *output*. Our metaphor is a

car. The e-Leaf can drive a reaction in two opposite directions—oxidation and reduction (like the forward and reverse gears of a car). The reaction rate (the *output*) can be increased, decreased or zeroed (the accelerator and brake) by selecting the appropriate electrode potential (the *input*). The overall rate of the process is observed as current (the speedometer), and integrating the current with respect to time gives the charge passed (the odometer)—producing a continuous read-out of the progress of a synthesis. The high degree of interactive control, allowing easy navigation across complex cascade reactions (the map) is not found for other enzyme cascade systems[9,18,53] which rely on entry of reactants, or light as inputs. Other electrochemically driven cascades, including 'one-pot' examples, depend also on artificial electron mediators[54,55].

## Misunderstandings

Development of the e-Leaf met with early misunderstanding, a likely consequence of its interdisciplinary nature. A common issue arose from the fact that instead of calculating the rate of catalysis from a time course, as is conventionally the case, the rate itself is displayed as a direct output, i.e. current = rate. A reviewer confused the time course for growth of activity (the rise in catalytic current to reach a limiting value as the second enzyme entered the pores) with a concentration-time plot (such as NADP(H) consumption), thereby concluding, incorrectly, that the catalytic reaction was slow and short-lived and that such a system could not be useful. The error was taken up with the editor who quickly realised that the reviewer had misunderstood the time-course—not realising that a flat line reached in an electrocatalytic time-course signifies that a steady-state has been achieved.

Another issue was a failure to recognise that the e-Leaf requires at least two enzymes, i.e. E1 and E2. In a computational paper on the merits of nanoconfinement of enzymes, it was concluded that nanoconfinement was a 'misleading' term after modelling only the case of E1 confined alone in which the enzyme electrocatalytically converted NADP(H) presented as a reactant contained in (and arriving from) the bulk solution. The authors ignored the fact that NADP(H) is an exchangeable cofactor that is rapidly recycled during catalysis—it is not a reactant that is consumed[48], and the second enzyme E2 must be present (making up the minimal pair) together with its substrates[56]. As shown in the next section, the nanoconfinement advantage is obtained only if E1 is coupled to E2 by *local* NADP(H) recycling—nanoconfinement limits the escape of cofactor and subsequent intermediates during the sequence of reactions. If only a single enzyme is loaded (which would need to be electroactive), the advantage of a mesoporous electrode is limited to increasing the coverage—i.e. scaling to improve electrochemical or spectroscopic detection. The e-Leaf corrals and organises the *collective* action of multiple enzymes: it is thus distinguished from earlier electrochemical/spectroelectrochemical work, such as the characterisation of cytochrome c molecules loaded at high concentration in mesoporous ITO[26].

## The ranges of NADP⁺/NADPH cycling

The e-Leaf highlights two interpretations of *range* in nicotinamide cofactor chemistry[57]: these observations have significance for how NAD(H) and NADP(H) are used in biotechnology, and their roles as central agents in cell metabolism[58].

The first aspect concerns the *thermodynamic* range of cofactor recycling, which encompasses its primary electrocatalytic regeneration and its secondary exchange in a dehydrogenase-catalysed reaction. The free energy profile shown in Fig. 4a addresses the sequence of steps in the reducing direction, viewed in terms of increasing (positive) formal potential. The direction and spontaneity of each step depend on how the (Nernstian) potential windows of donor and acceptor partners align and overlap. In step 1, two electrons transfer from the electrode to the FAD of FNR in a long-range tunnelling process[59] for which the electrode provides a continuous and adjustable energy source. For rapid electron tunnelling, FNR must bind in a productive orientation to the nanoparticle surface to achieve strong electronic coupling between ITO and FAD, analogous to the complex that forms with ferredoxin in the physiological situation[28,29]. At the same time, this interaction must allow enough mobility for NADP(H) to enter and

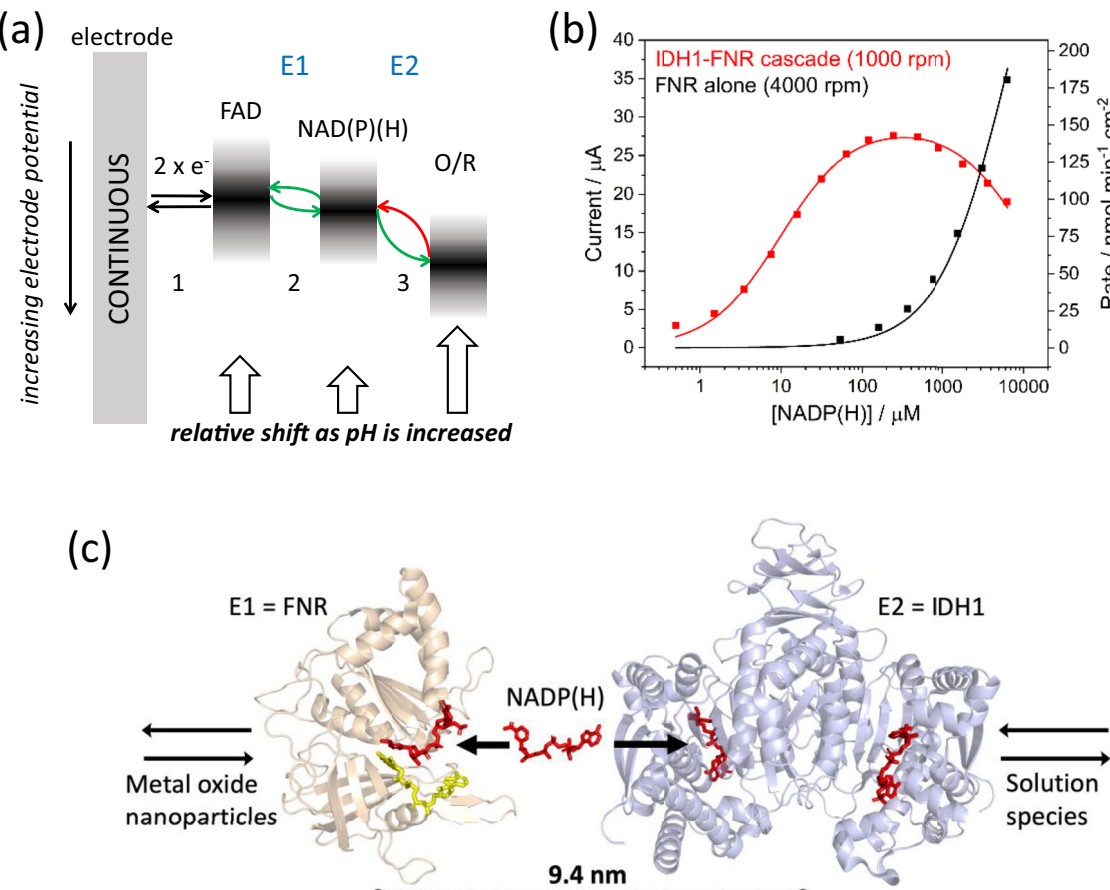

**Fig. 4 | The thermodynamic and spatial ranges of nicotinamide cofactor recycling. a** A free energy profile for electron/hydride transfer from an electrode to a reactant undergoing catalytic reduction at dehydrogenase E2. Transfers indicated in green are favourable, and those in red are unfavourable. **b** The >100-fold nanoconfinement advantage, showing the current measured at increasing concentrations of NADP(H) in solution. The black curve shows results with FNR (E1) and isocitrate dehydrogenase (E2) loaded but without isocitrate in the solution. The current depends on the FNR-catalysed oxidation of NADPH entering the ITO layer from the solution. The red trace shows results with the two-enzyme cascade reaction activated by adding isocitrate (10 mM): the current now depends on the rate of localised NADP(H) recycling. A rotating disc electrode, held at +0.2 V vs SHE, was used in each case. The data focus on conditions where NADPH < 10 mM to emphasise the regime that shows the largest enhancement. **c** Localised recycling between crowded enzymes: FNR, IDH1, and NADP(H) drawn to-scale with an FNR-IDH1 mean centre-to-centre distance of 9.4 Å, equivalent to a combined concentration of 2 mM (see text). Panel **b** was adapted with permission[41].

leave. In step 2, a hydride is transferred from FAD to $NADP^+$ in a reaction that is only slightly more favourable than the reverse transfer, since the potential window of the FAD overlaps that of $NADP^+$ throughout the pH range 5-9[27,32]. In step 3, catalysed by dehydrogenase E2, a hydride is transferred from NADPH to O (oxidised species), in this case in a unidirectional manner because the alignment and overlap strongly favour the reducing direction. The relatively narrow Nernstian window of a two-electron (vs. one-electron) transfer limits the options for the close alignment of potentials needed to permit bidirectional electrocatalysis, but pH plays an important role. Of the three steps along the chain, O/R ((oxidised/reduced species), is the most sensitive to pH (a $2e^-/2H^+$ reaction having a potential dependence of $-0.06$ V/pH unit is most common for organic molecules) so O becomes easier to reduce as the pH is lowered. In the terminology used to discuss catalytic bias in PFE, NADP(H) becomes the electrochemical control centre: it determines the catalytic potential and directionality of the catalytic O/R reaction[42,43]. We saw in Fig. 2d that the ketone/alcohol interconversion is unidirectional at pH 7.5 but bidirectional at pH 9.0, where the $NADP^+$/ NADPH and O/R potentials have become similar in value.

The second aspect concerns the *spatial* range of nicotinamide cofactor recycling. The advantage of local NADP(H) recycling was demonstrated by experiments carried out to compare how the cofactor is used, first as an exchangeable cofactor (in catalytic quantities) and then as a reactant. In these experiments, E2 was human isocitrate dehydrogenase I (IDH1) which

catalyses the oxidation of isocitrate to 2-oxoglutarate by $NADP^+$. The data are shown in Figure 4b[41]. Both E1 and E2 were loaded into the pores. With 10 mM isocitrate present in the solution (labelled as IDH1-FNR cascade), $NADP^+$ was titrated into the solution, whereupon the oxidation current increased steeply with each addition of increasing concentration, reaching a maximum value at just 0.2 mM. The solution was then exchanged, and the cell was thoroughly rinsed. The titration was repeated; however, this time, isocitrate was not added to the solution, and NADPH was used instead of $NADP^+$ in order to measure the electrochemical oxidation of NADPH catalysed by FNR present in the same electrode *without* nanoconfined NADP(H) recycling (i.e. measuring FNR activity for the bulk electrolysis of NADPH). This experiment is labelled 'FNR alone'. As NADPH was added, the oxidation current increased much more gradually compared to the condition in which NADP(H) was recycled within the electrode, approaching a plateau at >20 mM[41]. Notably, a greater than 100-fold catalytic advantage is achieved when NADP(H), an expensive component in biocatalysis[46], is recycled locally instead of being consumed as a reactant.

The advantage of localising NADP(H) was emphasised further by exploiting the observation that two molecules of NADP(H) are bound tightly to IDH1 in a resting state[41]. It was reasoned that IDH1 would carry this bound NADP(H) into the electrode, releasing it for recycling with FNR only when turnover was induced in the presence of isocitrate. The system, operating without any cofactor added to the solution, was active for five

# (a)  Interconversion between aspartate and pyruvate

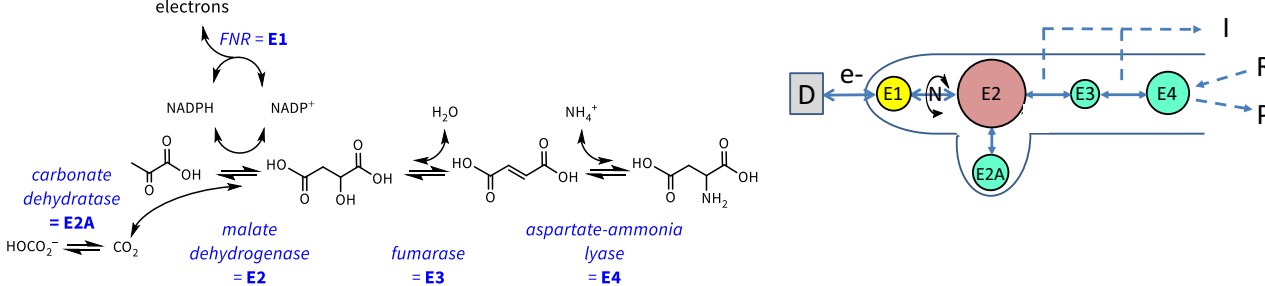

# (b)  Reduction of a carboxylic acid to its aldehyde

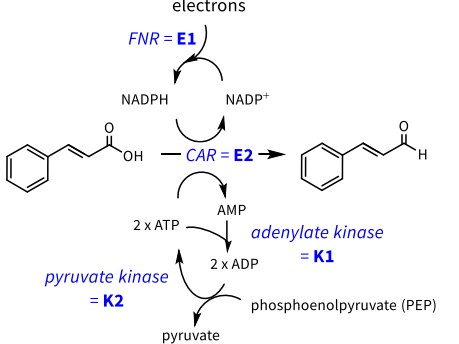

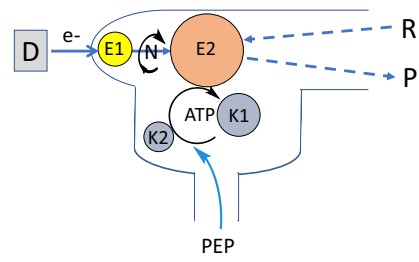

**Fig. 5 | Examples of extended enzyme cascades driven in the e-Leaf. a, b** Two examples of extended cascades, shown in conventional mode (*left*) to focus on the chemical conversions, and as cascade maps (*right*) to show the interrelationships between enzymes and the flow of information that is transduced by E1 into electrical current. The relative sizes of the enzyme circles emphasise the higher loading that is required to offset lower activity.

days, during which time the NADP(H) that had entered as cargo on an IDH1 molecule would undergo ~160,000 turnovers[41]. Although varying the enzyme loading ratios yielded different proportions of nanoconfined FNR and IDH1; in each case, the average combined enzyme concentration (i.e. [FNR] + [IDH1]) was approximately 2 mM when estimated using a few assumptions about ITO depth and void space[41]. At this concentration, the average enzyme centre-to-centre distance is only 9.4 nm[60,61]: based on this estimate alone, the minimal catalytic unit comprised of FNR, IDH1 and NADP(H) must resemble that shown in Fig. 4c.

The observation that NADP(H) recycling is highly localised raised the question of how effectively reactants (usually smaller than the cofactor) present in solution can reach the large proportion of the enzymes that must be deeply buried in the ITO layer. A computational study showed that although diffusion of small-molecule substrates is retarded when they have to pass through small channels, catalysis by an E2 enzyme that is deeply trapped still contributes strongly to the current when operating at typical conditions (<1 mA/cm$^2$)[56].

## Optimising cascade performance

As with its precursor technique PFE, the investigator holds a dialogue with an enzyme, but this dialogue could now be extended to enzymes of all major classes. This extension was made clear by two examples, the first being a five-enzyme cascade that catalyses the conversion of pyruvate, carbonate ions and ammonium ions into aspartate (Fig. 5a)[62]. In the second example, a carboxylic acid is reduced to its aldehyde in a reaction requiring ATP in addition to NADPH (Fig. 5b)[63].

In each case, the products (and intermediates) were identified and quantified by NMR. The performance of the cascade depends on the local concentration of enzymes and the ratios in which they are present, with less-

active enzymes being required in higher amounts to avoid bottlenecks. In Fig. 5, this empirical way to optimise the performance of a cascade is depicted by the size of the circle representing each component enzyme—a larger circle indicates that an enzyme is required in a higher quantity to compensate for its lower inherent activity. The overall capacity must ultimately be limited by how many enzyme molecules can physically be packed in the spaces[61,64]; accordingly, it was observed that activity is optimised at particular loading amounts and ratios. Unproductive escape of intermediates can be detected (if this occurs *upstream* of FNR) by rotating the electrode at high speed, which lowers the current if it enhances the escape of an intermediate, but raises the current if the supply of a reactant to the ITO surface is a limiting factor[62].

In Fig. 5a, co-entrapment of carbonic anhydrase (E2A) enabled in situ production of $CO_2$ for immediate use by malate dehydrogenase (E2)[62]. Figure 5b describes the electrocatalytic synthesis of cinnamaldehyde from cinnamic acid using carboxylic acid reductase (CAR, E2) and two kinases[63]. In this example, AMP, an inhibitory by-product of CAR, is rapidly sequestered by adenylate kinase (K1) and the subsequent regeneration of ATP from ADP by pyruvate kinase (K2) allows the cascade to operate at high efficiency when fuelled with a phosphate donor. By incorporating kinases into the e-Leaf, it is possible to drive reactions that, on the basis of Fig. 4a, would not be feasible using NAD(P)H alone because the hydride transfer (Step 3, the reduction of a carboxylic acid to an aldehyde) is too unfavourable. The CAR also acts as a branchpoint to connect pathways that use ATP and NAD(P)(H).

## Advantages and applications

The e-Leaf presents several important advantages over other ways of immobilising enzyme cascades. The hardware is straightforward, robust,

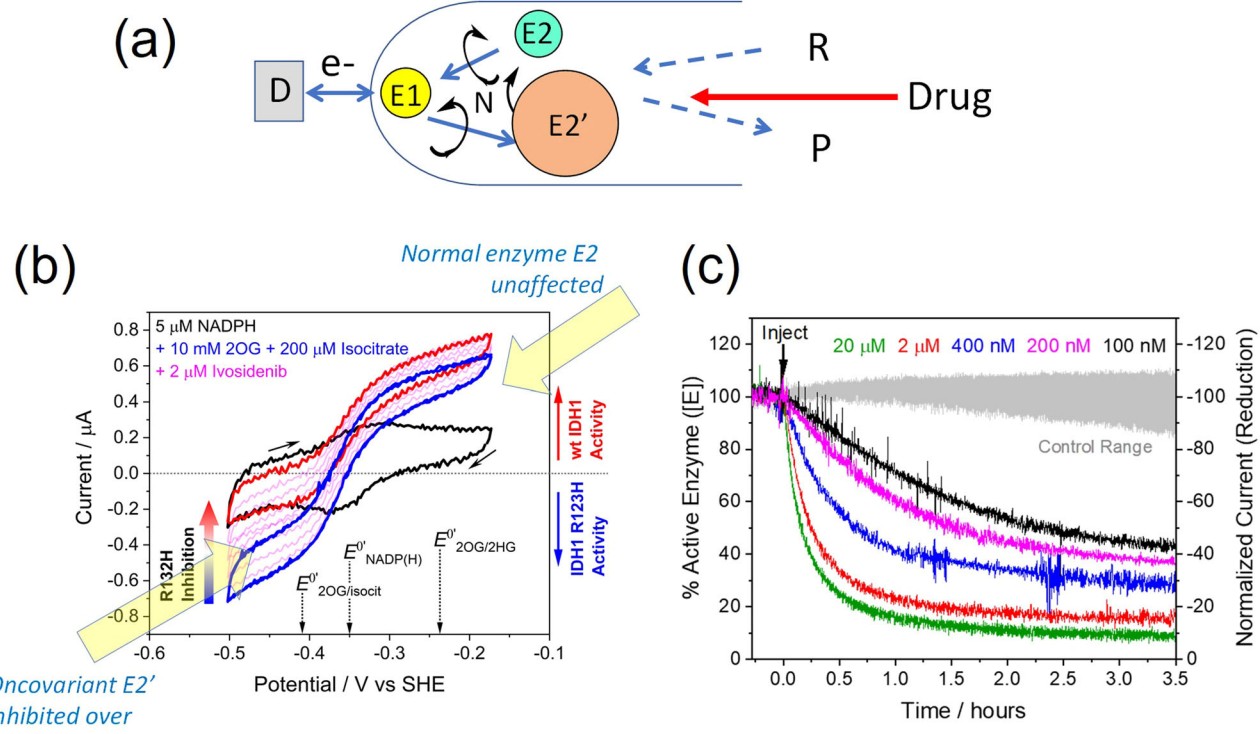

**Fig. 6 | The e-Leaf in drug mechanisms and development.** Applying the e-Leaf to study the selective inhibition of a single-site variant enzyme having undesired gain-of-function activity, in this case, human isocitrate dehydrogenase I (IDH1). **a** Cascade map depicting oxidation catalysis by E2 (native IDH1) and reduction catalysis by E2' (IDH1 R132H). The sizes of the spheres represent inverse relative activity, IDH1 R132H being the least active catalyst and required to be loaded in higher quantities. **b** The 'living' cyclic voltammogram (scan rate 1 mV/s) showing the simultaneous monitoring of two enzymes in a single experiment and how the

injection of a drug into the solution inhibits the conversion of 2-oxoglutarate to 2-hydroxyglutarate by E2' but does not inhibit the wildtype activity of E2. **c** The kinetics of the inhibition process can be studied under pseudo-first-order conditions, even with very low levels of the drug. A full analysis of the kinetics has been published: the results reveal how the drug first binds without inhibiting, then locks into its inhibitory position in a slow subsequent step. Panels **b** and **c** were adapted with permission from reference[68].

and can be scaled up (larger electrodes) or multiplexed (very small electrodes). Preparation is simple—a layer of nanoparticles is easily formed to a desired depth depending on deposition time, and enzymes, applied to the surface, enter the porous network spontaneously.

The benefits of nanoconfinement in terms of efficient cofactor recycling and channelling of intermediates are easily acquired. The technology is interactive, as described above in reference to the Dashboard, allowing an operator to communicate with enzyme cascades in dialogue mode. In contrast, an enzyme cascade trapped inside a freely diffusing colloidal-type material is relatively invisible—the reactions can be activated by chemicals or light, but neither of these can easily be made interactive or bidirectional. The rate and extent of reaction are not monitored directly, as they are when an electrical current is produced. The porous nature of the electrode allows a massive increase in enzyme loading, up to hundreds of monolayer equivalents, which is not achievable using conventional flat electrodes (e.g. graphite, or Au with a self-assembled monolayer). Another configuration, redox polymers, has also been reported for entrapping enzymes[65], but a crucial difference to the e-Leaf is that with a redox polymer, the potential is still governed by that of the polymer, which acts as an electron mediator, whereas the ITO electrode in the e-Leaf has metallic (electronically conductive) behaviour. As an example, the e-Leaf's ability to drive a process in each direction, forward or reverse, was exploited to deracemize a racemic secondary alcohol into its *R*- or *S*-forms using two alcohol dehydrogenases, one selective for *S*-, the other selective for *R*-, the latter being intentionally inactivated midway through an oxidation-reduction cycle[66].

The e-Leaf offers a way to synthesise or modify complex chemicals in a single operation[39,49,66,67]. In principle, the process involves identifying the enzymes required to convert A into Z, loading them in an optimal ratio into

the electrode nanopores, and energising the reaction via FNR and an adjacent dehydrogenase. Expensive nicotinamide cofactors are only required in catalytic amounts, with turnover numbers >1000 being easily achievable; and where required, ATP is supplied in situ by loading relevant kinases and adding a phosphate donor to the solution. An immense library of enzymes is available. Scaling up is assisted by depositing the oxide film on both sides of the support—multiples of flexible titanium foil cut to size and folded into shape[67]. Such an application is probably the most obvious one for exploitation, but the advantages may only surface for very high value-added compounds, ranging from pharmaceuticals to chemicals labelled both regio- and enantio-selectively with isotopes.

A very different application lies in enzyme and drug development: here, the advance lies in exploiting the relationship between steady-state rate and current, specifically, the fact that the electrocatalytic current is equivalent to the first derivative of a conventional solution (*concentration* vs time) trace—a consequence being that any reaction that alters the activity of an enzyme is detected directly as a *rate* vs time dependence. An example is the determination of the kinetic mechanism of a high-profile chemotherapy drug having a low IC$_{50}$ (low dose/high affinity)[68]. Although IDH enzymes normally catalyse the oxidation of isocitrate to 2-oxoglutarate by NAD(P)$^+$, certain mutations associated with cancers result in a gain-of-function activity—catalysis of the reduction of 2-oxoglutarate to 2-hydroxyglutarate by NAD(P)H[69,70]. The IDH1 R132H variant is inhibited by a drug known as Ivosidenib[71], which is in clinical use.

In an unusually illuminating electrochemical experiment, wildtype IDH1 (E2) and the variant (E2') were loaded into the same electrode (Fig. 6a), with the variant being present at a higher fraction to compensate for its much lower activity. Cyclic voltammetry (Fig. 6b) shows how the

e-Leaf enables activities of both enzymes—wildtype (oxidation) and variant (reduction)—to be monitored simultaneously in a single experiment. The addition of a low concentration of Ivosidenib causes selective inhibition of the R132H variant while the wildtype enzyme is left unscathed. The current on the reducing side of the cyclic voltammogram (2-oxoglutarate → 2-hydroxyglutarate) decreases with each cycle whereas that on the oxidising side (isocitrate → 2-oxoglutarate) actually increases slightly: the latter effect arises (see arrow connecting E2 and E2' in Fig. 6a) because as E2' is inhibited, more NADPH becomes available to be oxidised by the electrochemical (E1) pathway (Fig. 6b).

Although the local concentration of enzymes in the pores is very high, the actual quantity of enzyme can be extremely small, picomoles for a very small electrode, enabling the kinetics of action of an inhibitor to be studied very efficiently and in considerable detail under pseudo-first-order conditions, even at nanomolar solution concentrations[68]. It thus becomes straightforward to study slow-acting, high-affinity drugs at levels that would normally be too low to yield pseudo-first-order kinetics and even below the equivalent concentration that the enzyme would have, were the same quantity to be dispersed in solution. Ivosidenib is a slow-acting inhibitor, and monitoring reactions over several hours showed that the kinetics are first order for greater than two half-lives (Fig. 6c), a result that is mechanistically significant since IDH1 is a homodimer (Fig. 4c)—the action of a single Ivosidenib molecule binding at the dimer interface depending on the status of each monomer. The kinetic data showed that the Ivosidenib molecule must bind first in a rapid non-inhibitory manner before the actual inhibition step occurs in a subsequent slow reaction[68]. This application of the e-Leaf compares favourably with the use of surface plasmon resonance (SPR) to detect the binding of drugs (these may be in trace amounts) to a biological receptor[72–75]. The e-Leaf quantifies the inhibitory action of a potential drug, whereas SPR reports only on binding; details on the true inhibitory step may thus be obscured.

## Summary and outlook

The e-Leaf works for the following reasons.

- Enzymes are taken up spontaneously into a random mesoporous electrode material, typically to a depth of up to 3-6 microns. They retain their inherent catalytic properties (although collective properties may be modified).
- Enzymes of all major classes and covering a wide range of sizes become concentrated in the pores; they are nanoconfined, crowded, and may be in close contact.
- One of the enzymes, E1, acts as a transducer: it is endowed by nature with the ability for fast electron tunnelling. Ferredoxin-NADP$^+$ reductase (FNR) electrocatalytically cycles NADP(H) or NAD(H) cofactors: this is a role it performs in green plants, hence the term 'electrochemical Leaf'—e-Leaf. The transformed cofactor is used by the second enzyme E2, which recycles the cofactor and catalyses the next stage of the sequence.
- Unlike the chloroplast, however, the NADP(H) recycling may be driven in either direction: since FNR is a reversible electrocatalyst, an exquisite degree of electrochemical control is possible - the response time and information being uncorrupted by any requirement for promiscuous electron mediators. In their reduced states, the latter tend also to react rapidly with $O_2$, whereas NAD(P)H is unaffected.
- Enzymes have such high selectivities and efficiencies that, despite being randomly arranged in the pores, each intermediate along the reaction sequence is more likely to be processed in the next step than to escape the pores. The tight channelling renders NAD(P)(H) and intermediates efficient carriers of current and information, analogous to electrons and ions.

It would be challenging to mimic the e-Leaf using tandem systems based not on enzymes but on small molecular electrocatalysts and catalysts: they would need to be covalently anchored, and their generally lower activity

and selectivity would make a confined sequence of reactions much more difficult to achieve.

The e-Leaf can provide additional insight into the question of how enzymes function under very crowded conditions, as they may experience in a living cell[64,76–79]. At millimolar concentrations, centre-to-centre distances will be <12 nm, meaning enzymes are frequently in close contact (Fig. 4c)[60]. It is unclear yet how enzymes confined in such an environment (soft matter buried within a mineral network) might be physically observed and studied. The significance of Michaelis-Menten parameters is altered; for instance, the *local* concentration of the product of E2 in an extended cascade is probably well above its $K_M$ for the next enzyme E3. While the e-Leaf demonstrates the kinetic advantage of confining NAD(P)(H) cofactors, the degree to which the action of nicotinamide cofactors in vivo is localised or dispersed is of considerable interest[57,80]. Historically, central NAD(H) and NADP(H) pools have normally been assumed - the NAD(P)$^+$/NAD(P)H ratio in cells is a measure of metabolic state.

The e-Leaf helps us visualise the thermodynamics that drives enzyme cascades. It is rarely easy to explain or predict the course of complex multistep processes, but voltammetry comes to our aid—revealing, through inspection of catalytic currents and potentials along with pH dependence and even electrode rotation rate, a practical picture of the overall reaction landscape.

The concept of 'cascadetronics' represents an intriguing opportunity for development, as components can be chosen from an immense library of enzymes from all major classes to perform different interactive functions—feedback, sensing, potential gating—each of which could be processed by an analyser to facilitate enzyme or drug development. We have already mentioned how CAR provides a branchpoint between ATP and NAD(P)(H) pathways, rendering kinases electroactive[63]. Proteins, apart from enzymes, could be included if they influence catalysis in response to a stimulus. Additionally, while small molecule drugs remain relevant, new drug modalities such as peptides and peptidomimetics could also be monitored.

In summary, the e-Leaf is a concept that may be pursued in several different directions, each with myriad permutations due to the immense number of enzymes that are available to plug into the system. As a technology, it is scalable, and may be useful for synthesising high-value products, such as pharmaceuticals. At the other end of the scale, microelectrode versions should be well suited for multiplexing, leading to computer-controlled high-throughput assays for enzyme and drug development.

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

## Acknowledgements

Research on the e-Leaf has been supported by grants from the Biotechnology and Biological Sciences Research Council (BB/P023797/1) and the EPA Cephalosporin Fund (CF 327, 342 and 397). Important collaborations, particularly with Nicholas Turner and Christopher Schofield, have helped develop the science. FAA thanks St John's College, Oxford, for an Emeritus Research Fellowship and Hong Kong University for a Mok Hing-Yiu Distinguished Visiting Professorship.

## Author contributions

The first three authors are listed in the chronological order in which they carried out research to investigate and develop the e-Leaf in the laboratory of the corresponding author.

## Competing interests

The authors declare no competing interests.
