## [Peer Review File · Communications Chemistry]

REVIEWERS' COMMENTS:

Reviewer #1 (Remarks to the Author):

The manuscript discusses a novel biocatalysis platform - the electrochemical leaf (e-leaf). The authors present the principle of the novel approach, its advantages, applications already realized in the field, as well as its potential. The provided perspective is undoubtedly of high interest to researchers working in the field of enzymatic catalysis, but it also holds broader implications for other areas of research, such as biotechnology and medicinal chemistry. The text is logically written and is easy and enjoyable to read. There are only a few issues that I would suggest modifying to increase clarity:
Fig 2(c): Please define the abbreviations used for the dehydrogenase names shown in the figure in the figure caption.

Line 143: Please remove - after E1 in $E1 \leftrightarrow E2$.

Fig. 4(a): I found the presented scheme quite confusing. First, I would suggest removing the black arrow showing the direction of the electrode potential. Instead, a simple line with "+" and "-" or an arrow pointing in the opposite direction would be clearer. White arrows are used to indicate relative shifts of the redox couple with increasing pH. I could not understand the meaning of the white arrows pointing in both directions within the electrode.

Line 289: When describing Fig 4(b), it is noted that a plateau is reached at >20 mM, but these data points are missing from the figure. Additionally, the legend within the figure seems to be incorrect as it states that black data points correspond to FNR alone, whereas, in the text and figure caption, it is stated that black data points correspond to the cascade without isocitrate and red to the cascade with isocitrate.

Reviewer #2 (Remarks to the Author):

In the manuscript "Interactive biocatalysis – driving enzymes cascades into conductive material" the authors discuss the working principle of the "electronic leaf" obtained trapping different enzymes in a porous metal-oxide nanoparticle-based material. The manuscript provides an interesting reading on the steps that led to defining the bioelectrocatalytic processes taking place in the e-Leaf, and is of interest for scientists (both expert and non-expert) developing multi-enzyme biohybrid systems. Accordingly, the manuscript is recommended for publication after a minor revision based on the specific comments below.

Major comments:

It is stated that enzymes of all classes can be trapped in the e-Leaf and rendered electroactive, however this statement seems too optimistic. Specifically, if enzymes with their redox active sites buried deeply inside the protein structure are utilized, the presented approach might not be sufficient to ensure the "electrochemical wiring" of the enzyme, and redox mediators might be required. The sentence should be revised considering this aspect.

Further remarking this aspect, in the section "Achieving enzyme nanoconfinement with a simple material" it is stated that the random arrangement of NP allows trapping enzymes of different dimension, but again, having a different distribution of the nanostructured materials would most likely not allow the "electronic wiring" of all enzymes.

On page 3, in the section "Development of the e-Leaf" it is stated that "The separation between oxidation and reduction peaks was small and the narrow line-shape was as expected for a two-electron transfer having a high (but not total) degree of cooperativity." It would be beneficial to

expand the discussion on the peak separation, providing more details to the reader.

On page 4 the concept of "electrochemical reversibility" is briefly introduced, but it would be beneficial to improve the discussion of this topic, to better describe the concept to the reader.

On page 10 it is stated that "productive orientation to the nanoparticle surface is needed", since this is a critical aspect, discussing this point with details on the limitations of the electron transfer process between biological and non biological components as discussed in the semi-classical Marcus Theory of electron transfer would be a nice addition to the manuscript.

In the "Advantage and applications" a comparison of the e-Leaf advantages over freely diffusing colloidal material is presented. Here, it would be useful to compare the use of redox polymers to entrap enzyme-cascades and the approach of the e-leaf. This would allow a more balanced comparison of the e-Leaf to other available approaches.

REVIEWERS' COMMENTS:

Reviewer #1 (Remarks to the Author):

The manuscript discusses a novel biocatalysis platform - the electrochemical leaf (e-leaf). The authors

present the principle of the novel approach, its advantages, applications already realized in the field, as

well as its potential. The provided perspective is undoubtedly of high interest to researchers working in

ST. JOHNS COLLEGE

Oxford. OX1 3JP

Registered Charity 1139733

ST JOHN'S COLLEGE

OXFORD OX1 3JP

the field of enzymatic catalysis, but it also holds broader implications for other areas of research, such

as biotechnology and medicinal chemistry. The text is logically written and is easy and enjoyable to read. There are only a few issues that I would suggest modifying to increase clarity:

Fig 2(c): Please define the abbreviations used for the dehydrogenase names shown in the figure in the

figure caption.

Our response: The abbreviations have been defined in the legend. 'ADH = alcohol dehydrogenase, RedAm =

reductive aminase, ME = malic enzyme (malate dehydrogenase), (S)-IRED = (S)-imine reductase.

Line 143: Please remove - after E1 in $E1 \leftrightarrow E2$.

Our response: The typo has been removed.

Fig. 4(a): I found the presented scheme quite confusing. First, I would suggest removing the black arrow showing the direction of the electrode potential. Instead, a simple line with "+" and "-" or an arrow pointing in the opposite direction would be clearer. White arrows are used to indicate relative shifts of the redox couple with increasing pH. I could not understand the meaning of the white arrows

pointing in both directions within the electrode.

Our response: Figure 4(a) has been redrawn in accordance with these suggestions.

Line 289: When describing Fig 4(b), it is noted that a plateau is reached at >20 mM, but these data points are missing from the figure. Additionally, the legend within the figure seems to be incorrect as it

states that black data points correspond to FNR alone, whereas, in the text and figure caption, it is stated that black data points correspond to the cascade without isocitrate and red to the cascade with

isocitrate.

Our response: We have rewritten the section (now on pages 10-11) to help clarify these points:

.....first

as an exchangeable cofactor (in catalytic quantities) and then as a reactant. In these experiments, E2 was human isocitrate

dehydrogenase I (IDH1) which catalyses the oxidation of isocitrate to 2-oxoglutarate by NADP+

. The data are shown in

Figure 4b41

. Both E1 and E2 were loaded into the pores. With 10 mM isocitrate present in solution (labelled as IDH1-FNR

cascade), NADP+ was titrated into the solution whereupon the oxidation current increased steeply with each addition of

increasing concentration, reaching a maximum value at just 0.2 mM. The solution was then exchanged, and the cell was

thoroughly rinsed. The titration was repeated, however, this time, isocitrate was not added to the solution and NADPH was

used instead of NADP+

in order to measure the electrochemical oxidation of NADPH catalysed by FNR present in the same electrode without nanoconfined NADP(H) recycling (i.e. measuring FNR activity for the bulk electrolysis of NADPH). This

experiment is labelled 'FNR alone'. As NADPH was added, the oxidation current increased much more gradually compared

to the condition in which NADP(H) was recycled within the electrode, approaching a plateau at > 20 mM41

.

And in the figure legend we write:

The data focus on conditions where NADPH < 10 mM to emphasize the regime that shows the largest enhancement.

Reviewer #2 (Remarks to the Author):

In the manuscript “Interactive biocatalysis – driving enzymes cascades into conductive material” the

authors discuss the working principle of the “electronic leaf” obtained trapping different enzymes in a

porous metal-oxide nanoparticle-based material. The manuscript provides an interesting reading on the

steps that led to defining the bioelectrocatalytic processes taking place in the e-Leaf, and is of interest

for scientists (both expert and non-expert) developing multi-enzyme biohybrid systems.

Accordingly, the

manuscript is recommended for publication after a minor revision based on the specific comments below.

Major comments:

It is stated that enzymes of all classes can be trapped in the e-Leaf and rendered electroactive, however

this statement seems too optimistic. Specifically, if enzymes with their redox active sites buried deeply

inside the protein structure are utilized, the presented approach might not be sufficient to ensure the

“electrochemical wiring” of the enzyme, and redox mediators might be required. The sentence should be

revised considering this aspect.

Our response: The Reviewer may be misunderstanding the concept (see the next comment). The enzymes are rendered ‘electroactive’, not through direct electron transfer but because the whole sequence of reactions from FNR to E2, E3 etc including cofactor recycling is channeled by nanoconfinement and produces the current. For example, we would detect a perturbation at E3 (inhibition or activation) as a change in current. On page 2, we have added ‘and, as explained in this Perspective, rendered effectively ‘electroactive’: they do not need to possess a redox-active centre.’ Further remarking this aspect, in the section “Achieving enzyme nanoconfinement with a simple

material” it is stated that the random arrangement of NP allows trapping enzymes of different dimension, but again, having a different distribution of the nanostructured materials would most likely

not allow the “electronic wiring” of all enzymes.

Our response: The Reviewer has misunderstood the concept. Only one type of enzyme (E1, i.e. FNR)

needs to transfer electrons.

On page 3, in the section “Development of the e-Leaf” it is stated that “The separation between oxidation and reduction peaks was small and the narrow line-shape was as expected for a two-electron

transfer having a high (but not total) degree of cooperativity.” It would be beneficial to expand the discussion on the peak separation, providing more details to the reader.

Our response: We have clarified this section to explain the shape of the peaks and the interpretation of

the signal. On pages 3-4 we write ‘The narrow line-shape was as expected³⁰ for a two-electron transfer having a high

(but not total) degree of cooperativity, and the small separation between oxidation and reduction peaks implied fast electron

exchange. In the ideal reversible case, the cyclic voltammogram for a surface-immobilised redox couple consists of a pair of

gaussian-like peaks corresponding to reduction and oxidation – each maximising at the same electrode potential and

(provided all the molecules experience the same local environment) displaying a half-height width of approximately $90/n_{\text{eff}}$

mV at 25 °C. The term n_{eff} is the ‘effective electron number’ – the number of electrons transferred in an apparently

simultaneous (cooperative) process³¹. For a single electron transfer, n_{eff} is automatically equal to 1.0, but for a two-electron

reaction n_{eff} depends on the separation between the component one-electron potentials. If the second electron transfers at a

much more favourable potential than the first ($E_2 \gg E_1$), the one-electron intermediate is very unstable and $n_{\text{eff}} = 2$ (the

limiting case for a fully cooperative two-electron transfer). The peak height varies as n_{eff}^2

, making such a cooperative transfer

much easier to observe. The partially cooperative nature of the FNR signal ($1 < n_{\text{eff}} < 2$) was significant because in the

chloroplast, FNR receives two sequential one-electron transfers from separate ferredoxin molecules – requiring the oneelectron

On page 4 the concept of “electrochemical reversibility” is briefly introduced, but it would be beneficial

to improve the discussion of this topic, to better describe the concept to the reader.

Our response: We have clarified our use of this term: On page 4, the relevant section now reads ‘The term

‘reversible’ is often used casually to describe a reaction that is simply bidirectional, i.e. can be driven in either

direction: the electrochemical (thermodynamic) definition of reversibility is far stricter – it refers to a bidirectional

reaction for which the direction and rate respond fluently to a miniscule change in potential across the formal

potential value, i.e. only a very small overpotential (driving force) is needed^{36,38}

. Electrochemical reversibility is a

marker for efficiency – it signifies that the energy input is used to conserve thermodynamic requirements rather

than overcome kinetic barriers.

On page 10 it is stated that “productive orientation to the nanoparticle surface is needed”, since this is

a critical aspect, discussing this point with details on the limitations of the electron transfer process

between biological and non biological components as discussed in the semi-classical Marcus Theory of

electron transfer would be a nice addition to the manuscript.

Our response: We have elaborated on the interaction between FNR and nanoparticle surface. On page

10, we have written ‘For rapid electron tunnelling, FNR must bind in a productive orientation to the nanoparticle surface

to achieve strong electronic coupling between ITO and FAD, analogous to the complex that forms with ferredoxin in the

physiological situation^{28,29}

. At the same time, this interaction must allow enough mobility for NADP(H) to enter and leave.'

In the "Advantage and applications" a comparison of the e-Leaf advantages over freely diffusing colloidal material is presented. Here, it would be useful to compare the use of redox polymers to entrap

enzyme-cascades and the approach of the e-leaf. This would allow a more balanced comparison of the

e-Leaf to other available approaches.

Our response: We have added the following text to page 13. 'The porous nature of the electrode allows a massive

increase in enzyme loading, up to hundreds of monolayer equivalents, which is not achievable using conventional flat electrodes

(e.g. graphite, or Au with a self-assembled monolayer). Another configuration, redox polymers, has also been reported for

entrapping enzymes,⁶⁵ but a crucial difference to the e-Leaf is that with a redox polymer, the potential is still governed by that

of the polymer which acts as an electron mediator, whereas the ITO electrode in the e-Leaf has metallic (electronically

conductive) behaviour.

On page 15, we have also added 'Unlike the chloroplast, however, the NADP(H) recycling may be driven in either

direction: since FNR is a reversible electrocatalyst, an exquisite degree of electrochemical control is possible - the response

time and information being uncorrupted by any requirement for promiscuous electron mediators. In their reduced states, the

latter tend also to react rapidly with O₂, whereas NAD(P)H is unaffected.